# The Clinical Effects of Pixel CO_2_ Laser on Bladder Neck and Stress Urinary Incontinence

**DOI:** 10.3390/jcm11174971

**Published:** 2022-08-24

**Authors:** Cheng-Yu Long, Jennifer Po-Ning Lee, Zi-Xi Loo, Yi-Yin Liu, Chang-Lin Yeh, Chien-Wei Feng, Kun-Ling Lin

**Affiliations:** 1Department of Obstetrics and Gynecology, Kaohsiung Medical Hospital, Kaohsiung Medical University, Kaohsiung 80708, Taiwan; 2Department of Obstetrics and Gynecology, Kaohsiung Municipal Siao-Gang Hospital, Kaohsiung Medical University, Kaohsiung 80708, Taiwan; 3Regenerative Medicine and Cell Therapy Research Center, Kaohsiung Medical University, Kaohsiung 80708, Taiwan; 4Vaginal Rejuvenation & Regenerative Center, Vigor Aesthetic Medical Group, Kaohsiung 81358, Taiwan; 5Department of Obstetrics and Gynecology, Kaohsiung Municipal Ta-Tung Hospital, Kaohsiung Medical University, Kaohsiung 80145, Taiwan; 6Department of Obstetrics and Gynecology, College of Medicine, Kaohsiung Medical University, Kaohsiung 80708, Taiwan

**Keywords:** Pixel CO_2_ laser, bladder neck, stress urinary incontinence, perineal sonography

## Abstract

Background: Our study aims to assess Pixel CO_2_ laser efficacy for female stress urinary incontinence (SUI). Methods: In the study, 25 women with SUI were included and scheduled for vaginal Pixel CO_2_ Laser (FemiLift™, Alma Lasers, Israel) treatment. All subjects had a baseline and 6-month post-treatment assessment that included three-dimensional perineal ultrasound and validated questionnaires. Results: Data showed that monthly three-session vaginal Pixel CO_2_ Laser treatment significantly improved SUI symptoms, as evidenced by validated questionnaires, including UDI-6, IIQ-7, ICIQ, and vaginal laxity questionnaire (*p* < 0.05). The Pixel CO_2_ Laser efficacy in vaginal treatment was 20/25 (80%), and the perineal sonography showed that laser treatment significantly decreased bladder neck mobility and middle urethral area (during resting and straining). Permanent adverse events were not found. Conclusions: The results of our study suggested that for the treatment of mild to moderate SUI symptoms, Pixel CO_2_ Laser is effective and safe; however, more studies and a longer follow-up should be conducted to confirm its efficacy and durability.

## 1. Introduction

One of the gynecological diseases that cause urine leakage during an increase in abdominal pressure, such as coughing, sneezing, laughing, or exercising, is stress urinary incontinence (SUI). Previous studies reported that approximately 50% of women with UI report SUI as the primary or sole symptom of incontinence [1]. Other population research groups indicated that the prevalence of female SUI is 26.4% [2]. The multinational Epidemiology of Lower Urinary Tract Symptoms Study also reported SUI in 44% of women studied, depending on the presence of associated symptoms [3]. In Taiwan, in recent decades, one in three women have suffered from SUI [4]. When the muscles which support the bladder become weaker, SUI occurs, which leads to the bladder neck descent and inhibition of normal urethral function [5]. Obesity, pregnancy, vaginal delivery, diabetes mellitus, pelvic surgery, genetic factors, and advancing age are some factors involved in the etiology of SUI. Additionally, with increasing age, the loss of muscle fibers is replaced by scar tissue and the denervation of relevant pelvic floor structure [6].

In clinical practice, SUI treatments should always start with noninvasive methods such as weight loss, hormonal therapy, physiotherapy, pelvic floor exercise, or pessaries [7]. Surgical intervention is considered when these treatments fail to improve SUI symptoms within 3 to 6 months. Surgical treatment is the gold management standard, mainly tension-free sub-urethral slings [7]. The success rate of tension-free sub-urethral slings was reported in 80–95% of patients [8]. Barisiene et al., (2018) also showed that the total TVT and TOT complication rate in TVT and TOT was observed in 16.07% and 5.1% of patients, respectively [9].

Previous literature showed that pulsed laser photothermal energy might repair collagen structure and induce neo-collagenesis to the nearby tissue by skin [10,11] and pelvic floor [12]. The collagen fiber would be activated and contracted until the temperature reached 60 degrees. Neo-collagenesis, elastogenesis, and neo-angiogenesis are promoted by the increasing temperature [13]. Indeed, recently, transvaginal laser therapy has been used to treat SUI, with a variable cure rate from 21% to 38% [14]. Our team also demonstrated that the erbium:yttrium-aluminum-garnet (Er: YAG) vaginal laser might play a role in treating mild to moderate SUI symptoms, which is partially attributed to the improvement of bladder neck mobility and vaginal laxity [15,16]. However, some research revealed that CO_2_ laser also induces tissue tightening and that the fibroplasia per micrometer depth of damaged skin is more evident and lasts longer than Er: YAG laser [17,18]. Assessment of bladder neck mobility is a part of the evaluation of SUI. Transperineal ultrasonography has also been introduced to evaluate the mobility of the bladder neck. Many studies focused on the degree of urethral angle, posterior urethrovesical angle, bladder neck descent, and rotation angles [19,20]. Al-Saadi et al., 2016 demonstrated both angles were significantly different between the groups at rest and straining, and there was a significant upregulation in the value of each angle. Higher values of increment were reported in the SUI group [21]. Our previous study in 2012 examined the hypoechoic area of the proximal, middle, and distal urethra through perineal sonography used to compare the therapeutic effect of TVT and TVT-O in rest or strain [22].

Among the CO_2_ laser techniques, the Pixel CO_2_ Laser technology could pixelate the beam to either 49 or 81 pixels. The pixelated beams could generate microablative damage through these microscopic columns. The deep thermal effect of the Pixel applicator enables minimal ablation to the patient, which is ideal for SUI patients. A faster regeneration of the undamaged tissue is demonstrated when wound healing begins. Thus, to fully assess the therapeutic effect of the laser treatment, this study examined the effect of the novel Pixel CO_2_ Laser on SUI symptoms with SUI-related questionnaires, urodynamic tests, and urogenital topography by perineal ultrasound.

## 2. Materials and Methods

This prospective study included 25 patients (mean age 42.9 ± 5.6 years) suffering from SUI at the Department of Obstetrics and Gynecology, Kaohsiung Medical University Hospital. The exclusion criteria included stage 2 or higher based on the Pelvic Organ Prolapse Quantification system (POPQ) [23]. In this study, all patients had bladder neck hypomobility (urethral hypomobility, type 2 SUI), and none had intrinsic sphincter deficiency (type 3 SUI).

The treatment administered to patients included in this study was three sessions of vaginal fractional microablative Pixel CO_2_ Laser system (FemiLift™, Alma Lasers, Israel) via a handpiece probe. The treatment interval was 28 days, and the laser treatment was performed per the manufacturer’s instructions. Thus, every patient received different laser intensities as tolerated (70 to 120 mJ/px). Moreover, the laser beam was fractionated into 81 “px” per cm^2^ at each emission, and the expected depth of microablation varied between 300 and 500 μm with increased thermal margins of 150–200 μm. The patient was placed in a lithotomy position, a laser handpiece was inserted into the vaginal canal, and then the probe was rotated clockwise to cover the entire vaginal wall. This movement was repeated three times.

Before the treatments and 6 months after the last laser treatment, each patient’s baseline characteristic data were collected. A personal interview was conducted in the following battery of questionnaires: Overactive Bladder Symptom Score (OABSS) [24], Urogenital Distress Inventory 6 (UDI-6), Incontinence Impact Questionnaire 7 (IIQ-7) [25], Incontinence Questionnaire (ICIQ) [26], and Vaginal Laxity Questionnaire (VLQ) [27] to assess the effect of laser on SUI symptoms and the impact on their quality of life. The ICIQ scores are divided into the following four severity categories: slight (1–5), moderate (6–12), severe (13–18), and very severe (19–21) [28]. The tests used in the study are listed below. A positive pad test is defined as a pad weight > 1 g. The pad test procedure followed the standard protocol from ICS for a 1 h pad test [29]. According to the International Continence Society [30], urodynamic studies, including non-instrumented uroflowmetry, filling and voiding cystometry, and urethral pressure profilometry, was performed using a 6-channel urodynamic monitor (MMS; UD2000, Enschede, Netherlands). Any uninhibited detrusor contraction during filling cystometry was deemed positive for detrusor overactivity.

Before and 6 months after treatment, a three-dimensional (3-D) transperineal ultrasound was performed for urethral topography in women with SUI who received CO_2_ pixel vaginal laser treatment. The bladder neck mobility and urethral area were measured as in the previous study [31]. The Volusion General Electric Sonography 730 Expert (GE, Healthcare Ultrasound, Zipf, Austria) was used with a 3.5-MHz curved linear- array. The transducer was placed between the labia majora and underneath the external urethral orifice. The sagittal view measured bladder neck mobility at rest and during the Valsalva maneuver at baseline and 6 months after the third treatment (Figure 1). Tracing the outer border of the urethral area at rest and during the Valsalva maneuver using the 3-D mode, the hypoechoic area of the proximal, middle, and distal urethras were measured (Figure 2).

Data were represented as mean ± standard deviations, and a *p* < 0.05 indicates a statistically significant difference. Statistical analyses were performed using IBM SPSS Statistical Software version 20.0 ed. Paired *t*-tests were performed for two related units on a continuous outcome. A *p*-value of less than 0.05 indicates statistical significance.

## 3. Results

A total of 25 women were included in the study. SUI patient characteristics are shown in Table 1. The mean average age of patients was 42.9 ± 5.6 years, and one patient was menopausal. The pad test was performed before and 6 months after three sessions of monthly treatments using vaginal Pixel CO_2_ Laser. The ICIQ scores of all subjects were classified into mild (*n* = 12; 48%), moderate (*n* = 9; 36%), severe (*n* = 2; 8%), and very severe (*n* = 2; 8%). In the pad test, urine leakage decreased significantly, and the ICIQ data showed that 80% (20/25) reported improvement in SUI symptoms (*p* = 0.023) (Table 1). Two women with very severe SUI experienced no beneficial effect after laser intervention, and the remaining three women suffered from mild, moderate, and severe SUI before treatment.

The results revealed that OABSS scores did not change significantly from 3.5 ± 2.6 to 3.2 ± 2.0 after the laser treatment (*p* = 0.481) (Table 2). In the UDI-6 questionnaires, a significant decrease in scores from 22.2 ± 14.1 to 14.2 ± 11.5 (*p* = 0.012) after treatment was shown, and the IIQ-7 scores also significantly dropped from 12.6 ± 14.8 to 7.9 ± 11.4 (*p* = 0.049). The ICIQ data also significantly decreased from 5.9 ± 4.2 to 3.2 ± 3.5 (*p* = 0.022). Additionally, the VLQ scores were significantly higher from 3.6 ± 0.7 to 4.8 ± 0.9 (*p* = 0.004) (Table 2).

All urodynamic parameters did not show statistical significance (*p* > 0.05), including maximum flow rate, residual urine, bladder volume at first sensation to void, maximal cytometric capacity, detrusor pressure at peak flow, maximum urethral closure pressure, and functional urethral length (Table 3).

In the 3D perineal ultrasound sagittal view, bladder neck mobility during Valsalva decreased significantly from 1.52 ± 0.3 cm to 1.27 ± 0.3 cm after treatment. Following laser treatment, the 3-D perineal ultrasound showed a significant decrease in the middle urethral area from 2.8 ± 1.1 mm^2^ to 2.2 ± 0.7 mm^2^ at resting (*p* = 0.045) and 3.5 ± 1.2 mm^2^ to 3.2 ± 0.8 mm^2^ at straining. However, the proximal and distal urethra areas showed no significant difference (*p* > 0.05) (Table 4).

## 4. Discussion

Our current study showed that the vaginal Pixel CO_2_ Laser treatment performed thrice after 6 months significantly improved SUI symptoms, as evidenced by perineal ultrasound and validated questionnaires including UDI-6, IIQ-7, ICIQ, and VLQ. The ultrasound data showed that bladder neck mobility and proximal area significantly decreased after the laser intervention. Due to the severe impact of SUI on the quality of life, numerous treatment modalities were developed from conservative to surgical intervention, such as lifestyle modification, pelvic floor muscle training, and mid-urethral slings [32]. Although surgical methods have a high cure rate [33], in our series, 84% (21/25) experienced mild to moderate SUI, similar to our previous studies.

The photothermal heating from lasers could lead to collagen denaturation, enhancing collagen remodeling and neo-collagenesis in the treated tissue. With the neo-collagenesis stimulation at a temperature of 60 to 65℃, laser treatment could shrink collagen without destroying its fiber. In 2015, the first Er: YAG laser study showed that laser treatment significantly decreased ICIQ-UI SF scores [34]. They observed that the ICIQ-SF score changed from 12.0 to 4.0 6 months after treatment. Similarly, we found that the treatment of Pixel CO_2_ Laser significantly decreased the ICIQ score from 5.9 ± 4.2 to 3.2 ± 3.5. However, the laser effect seems to only benefit women with mild to moderate SUI and not very severe incontinence.

Tien et al., examined the therapeutic effect of the Er:YAG laser in SUI and observed that the weight of the pad test significantly decreased from 14.0 ± 18.2 to 3.1 ± 5.6 g, with an efficacy of 78% (44/56) [35]. In our study, the Pixel CO_2_ Laser decreased the pad weight from 4.3 to 0.7 g, and the efficacy was 80 % (20/25) 6 months after the first intervention. This also indicated a comparable efficacy of pixel CO_2_ laser with Er: YAG laser for SUI. Moreover, Tien et al., found that the treatment of the Er:YAG laser could improve LUTS, quality of life, and sexual function. However, similar to our study, the urodynamic parameters did not change after the treatment. A rationale could be that the urodynamic study may lack sensitivity in detecting such subtle changes.

The Er: YAG laser (SMOOTH, Fotona, Slovenia) is a non-ablative laser delivered monthly for three continuous months. Fistonić et al., (2016) showed that 34/47 (72.3%) participants experienced improvement from baseline until the 2nd to 6th month of follow-up. We also previously evaluated the Er:YAG laser effect on SUI symptoms [15] and obtained a similar efficacy of 75.5% (31/41) 6 months after the intervention. Isaza et al., (2018) found that fractional CO_2_ laser (MonaLisa Touch™, Deka, Florence, Italy) treatment was also associated with a significant improvement in ICIQ-UI SF scores and pad weight test, not only at 12-months but also at the 24-months and 36-months follow-up. The results were also confirmed by significant histological changes [36].

Although Pixel (fractional) CO_2_ laser technology patients had a lower rate of complications than non-fractionated ablative laser treatment, adverse effects could still occur with the best technology. Fractional CO_2_ lasers were usually indicated for aesthetic purposes such as post-acne scarring [37]. Hence, the most common complications of fractional CO_2_ laser reported were erythema (95.38%) and burning sensation (92.30%) after the procedure [38]. Some side effects, which include dryness and edema, were observed in approximately 70% of patients [39]. The necessary precautions should be employed during the procedure to avoid possible side effects. This study’s ultrasound findings also confirmed the beneficial effect of CO_2_ lasers on SUI symptoms, partly attributed to decreased bladder neck motility 6 months after the intervention. None of these cases reported obvious side effects.

Transperineal ultrasound (TPUS) has been used as a valuable, noninvasive, and easy operating tool for quantitative assessment of the severity of lower urinary tracts (LUTs) in women in these few decades [19,21,40]. The precise evaluation of stress incontinence is essential for preventing unnecessary medical-surgical interventions. Hajebrahimi et al., showed that the difference between SUI patients and healthy women showed not only in the ICIQ-SF questionary but also in the hypermobility of the urethra detected by TPUS [40]. Besides, our previous report in Er: YAG laser also used TPUS to assess the therapeutic effect of laser therapy. The results showed that the decrease in the middle urethral hypoechoic area is more evident, consistent with bladder neck mobility [15].

Some studies argue that SUI results from urethral hypermobility and intrinsic sphincter deficiency. Bladder neck mobility assessment by perineal ultrasound is integral to SUI evaluation. Many studies focused on the degree of urethral angle, posterior urethrovesical angle, bladder neck descent, and rotation angles [19,20]. Our recent study also demonstrated that Er:YAG laser treatment significantly decreased bladder neck mobility and middle urethral area (during rest and straining) [15]. Thus, the remodeling effect of both Er:YAG and Pixel CO_2_ Laser should be similar for the treatment of SUI. However, more cases and extended follow-up periods are needed to prove the hypothesis of Pixel CO_2_ Laser.

Our study showed that the vaginal Pixel CO_2_ Laser treatment performed thrice after 6 months significantly improved SUI symptoms, as evidenced by perineal ultrasound and validated questionnaires including UDI-6, IIQ-7, ICIQ, and VLQ. Although urodynamic parameters did not show a significant difference, data revealed that laser treatment improved urinary symptoms and increased patients’ quality of life. Furthermore, Pixel CO_2_ Laser treatment significantly decreased bladder neck mobility and middle urethral area at rest and during straining. Reviewing the literature, this is the first study exploring the SUI treatment mechanism of vaginal Pixel CO_2_ Laser by perineal ultrasound. More randomized trials should be performed to evaluate the safety and durability of this new technology to enhance its application in women with SUI or other indications.

## 5. Conclusions

This study demonstrated that the vaginal Pixel CO_2_ Laser treatment administrated thrice after 6 months significantly improved SUI symptoms, as evidenced by perineal ultrasound and validated questionnaires including UDI-6, IIQ-7, ICIQ, and VLQ. The data revealed that laser treatment not only improved urinary symptoms but also increased patients’ quality of life, although urodynamic parameters did not show significant differences. Additionally, we observed that Pixel CO_2_ Laser treatment significantly decreased bladder neck mobility and middle urethral area (both at rest and during straining).

## Figures and Tables

**Figure 1 jcm-11-04971-f001:**
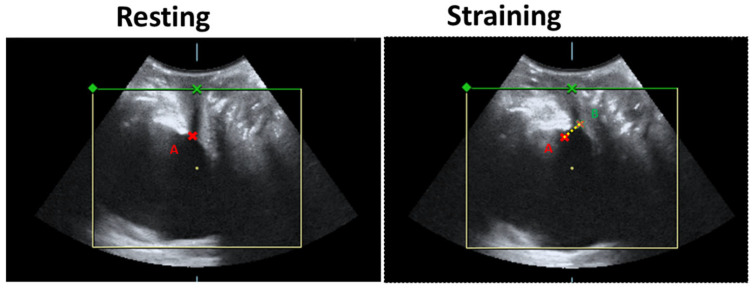
Bladder neck mobility from resting to straining. **A**, bladder neck at rest. **B**, bladder neck during straining.

**Figure 2 jcm-11-04971-f002:**
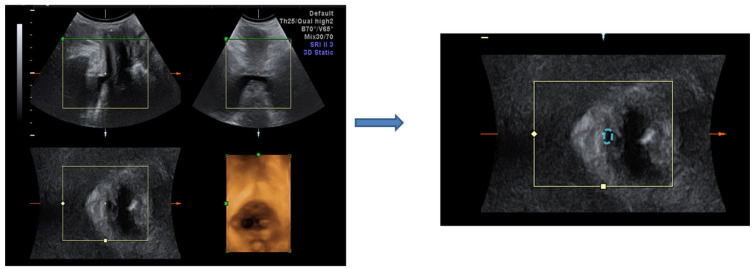
The urethral area is the hypoechoic area marked in a blue dotted line.

**Table 1 jcm-11-04971-t001:** Clinical background of the participants. Data are given as mean ± standard deviation or *n* (%).

	Pre-Operation (n = 25)	Post-Operation (n = 25)	*p* Value
**Mean age (years)**	42.9 ± 5.6		
**Mean BMI (kg/m^2^)**	22.6 ± 3.5		
**Menopause**	1 (4%)		
**Pad test**	4.3 ± 12.1	0.7 ± 1.3	0.023 *
**SUI grade by ICIQ**			
**mild**	12 (48%)		
**moderate**	9 (36%)		
**severe**	2 (8%)		
**very severe**	2 (8%)		
**Type of SUI (Type 2)**	25 (100%)		
**Efficacy**		20/25 (80%)	
**Follow-up (months)**		6M	

BMI, body mass index; SUI, stress urinary incontinence; ICIQ, International Consultation on Incontinence Questionnaire. * *p* < 0.05, Student’s *t*-test.

**Table 2 jcm-11-04971-t002:** Questionnaire results at baseline and 6 months post-treatment.

	Pre-Operation (n = 25)	Post-Operation (n = 25)	*p* Value *
**OABSS**	3.5 ± 2.6	3.2 ± 2.0	0.481
**UDI-6**	22.2 ± 14.1	14.2 ± 11.5	0.012 *
**IIQ-7**	12.6 ± 14.8	7.9 ± 11.4	0.049 *
**ICIQ-SF**	5.9 ± 4.2	3.2 ± 3.5	0.022 *
**VLQ**	3.6 ± 0.7	4.8 ± 0.9	0.004 *

OABSS, overactive bladder symptom score; UDI-6, Urogenital Distress Inventory; IIQ-7, Incontinence Impact Questionnaire; ICIQ-SF, Incontinence Questionnaire-Short Form; VLQ, Vaginal Laxity Questionnaire. Values are expressed as mean ± standard deviation or numbers. * Statistical significance; paired *t*-test.

**Table 3 jcm-11-04971-t003:** Urodynamic changes at baseline and 6 months post-treatment. Data are given as mean ± standard deviation.

	Pre-Operation (n = 25)	Post-Operation (n = 25)	*p* Value *
**Qmax (mL/s)**	24.7 ± 10.0	26.7 ± 8.8	0.290
**RU (mL)**	77.1 ± 85.6	53.1 ± 57.3	0.293
**Vfst (mL)**	202.9 ± 90.0	195.2 ± 86.6	0.73
**MCC (mL)**	444.6 ± 134.4	453.7 ± 155.7	0.530
**Pdet Qmax (mmHg)**	26.8 ± 15.8	27.4 ± 12.1	0.940
**MUCP (mmHg)**	68.3 ± 20.6	65.4 ± 19.1	0.486
**FUL (cm)**	30.5 ± 6.9	29.7 ± 4.5	0.631

Qmax, maximum flow rate; RU, residual urine; DO, detrusor overactivity; Vfst, first sensation to void; MCC, maximum cystometric capacity; Pdet Qmax, detrusor pressure at peak flow; MUCP, maximum urethral closure pressure; FUL, functional urethral length. Values are expressed as mean ± standard deviation or numbers. * Statistical significance; paired *t*-test.

**Table 4 jcm-11-04971-t004:** Urethral topography at baseline and 6 months after treatment during resting and straining.

		Pre-Operation (n = 25)	Post-Operation (n = 25)	*p* Value *
**Bladder neck mobility (mm)**		1.52 ± 0.3	1.27 ± 0.3	0.043 *
**Urethral area (mm^2^)** **Resting**	proximal	3.2 ± 1.8	2.7 ± 0.6	0.122
	middle	2.8 ± 1.1	2.2 ± 0.7	0.045 *
	distal	2.3 ± 1.0	2.5 ± 1.2	0.787
**Urethral area (mm^2^)** **Straining**	proximal	4.1 ± 1.3	4.2 ± 1.1	0.574
	middle	3.5 ± 1.2	3.2 ± 0.8	0.024 *
	distal	2.9 ± 1.0	3 ± 1.5	0.746

Values are expressed as mean ± standard deviation or numbers. * Statistical significance; paired *t*-test.

## Data Availability

The data presented in this study are available on request from the corresponding authors. The data are not publicly available due to privacy.

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
