# Peer review of "The Clinical Effects of Pixel CO_2_ Laser on Bladder Neck and Stress Urinary Incontinence"

_jcm, 2022, doi:10.3390/jcm11174971_

Round 1
Reviewer 1 Report
The investigation of the vaginal Pixel CO2 laser's SUI treatment mechanism by perineal ultrasound is presented in that work for the first time. The subject under discussion is intriguing.
The results of urinary questionnaires and objective indicators are frequently nonparametric data, in my experience. Please determine whether or not the study's data are parametric. The results may not dramatically change if the data are nonparametric, but it may be preferable to alter the test. I don't know a lot about statistics. Since there is, in my opinion, a wide range of views on statistics, I do not think it is necessary to change the data if peer review by a statistical expert—, not me—discovers no severe problems.
Author Response
Reviewer 1.
The investigation of the vaginal Pixel CO2 laser's SUI treatment mechanism by perineal ultrasound is presented in that work for the first time. The subject under discussion is intriguing.
Q1: The results of urinary questionnaires and objective indicators are frequently non-parametric data, in my experience. Please determine whether or not the study's data are parametric. The results may not dramatically change if the data are non-parametric, but it may be preferable to alter the test. I don't know a lot about statistics. Since there is, in my opinion, a wide range of views on statistics, I do not think it is necessary to change the data if peer review by a statistical expert—, not me—discovers no severe problems.
Our reply: Thanks for the valuable suggestion, we’ve checked some literature within two years and found that some studies used the same analysis process (pair t-test) as our study included [1], Lo et al., 2020 demonstrated that the difference of UDI-6, IIQ-7, ISQ-7, Qmax, RU and MUCP values between pre- and post-treatment group were analyzed by parametric tests [1]. Wang et al., 2021 also compared the difference between pre- and post-treated UDI-6 and OABSS scores by pair T-tests [2]. However, Wen et al., 2020 conducted a three-arm, non-inferiority, multi-center randomized controlled clinical trial related to moderate and severe overactive bladder and analyzed their outcome in both covariance analysis and non-parametric tests [3]. Thus, we suggested that these two analysis methods should be available in our study.
- Lo, T.S.; Jaili, S.; Tan, Y.L.; Wu, P.Y. Five-year follow-up study of Monarc transobturator tape for surgical treatment of primary stress urinary incontinence. Int Urogynecol J 2016, 27, 1653-1659, doi:10.1007/s00192-016-3019-3.
- Wang, S.; Wen, H.; Gao, Y.; Lv, Q.; Cao, T.; Wang, S.; Wang, J.; Li, Y.; Wang, H.; Wang, Z.; et al. Assessment of Pelvic Floor Function and Quality of Life in Patients Treated for Cervical Cancer: A Multicenter Retrospective Study. Gynecologic and obstetric investigation 2021, 86, 353-360, doi:10.1159/000517995.
- Wen, Q.; Li, N.; Wang, X.; Li, H.; Tian, F.; Chen, W.; Lu, Y.; Liu, Z. Effect of electroacupuncture versus solifenacin for moderate and severe overactive bladder: a multi-centre, randomized controlled trial study protocol. BMC complementary medicine and therapies 2020, 20, 224, doi:10.1186/s12906-020-03018-y.

Reviewer 2 Report
The current study focuses on improving the lower tract urinary symptoms and female incontinence through non-invasive approaches. By boosting the vaginal collagen regeneration rate, the suspensory system of the urethra is strengthened, with significant impact upon the quality-of-life of the patients and minimal discomfort.
Overall, the paper is well put together. Regarding the use of English language, I would suggest some minor revisions to be done, as it can be improved.
The introduction describes concisely the current therapeutic options for the chosen pathology, as well as the novelty that Pixel CO2 Laser can bring in the treatment scheme of stress urinary incontinence. For this section, I would suggest for the authors to offer a clear definition of ‘’pixel’’ laser, as well as mentioning the present success versus failure rates and complication rates of Tension-Free sub-urethral slings.
Regarding Materials and methods, the actual employed technique could be better and explained and further detailed. Additionally, intra-procedural pictures would be of great interest. Furthermore, this section could include a ‘definitions’ segment, where the authors should explain the urethral deviation and kinking angles and their normal values, as well as definitions for ‘positive pad test’ and if they measured the incontinence degree by assessing the number of used pads/day or by weighting each pad. Also, tables where each applied questionnaire could be detailed should be included.
The Results highlight the main significant findings in clear and concise manner. Table 1 could detail the causes of urinary incontinence in each case. Moreover, I recommend to include more follow-up data from the failed cases, the presumed reason and the next therapeutic option for these women.
Finally, the Discussions could benefit from adding a paragraph with the reported complications of Pixel CO2 laser in literature (Fibrosis? Cystitis?), as well as other uses for this type of laser therapy. My final comment concerns the fifth paragraph of this section. The presented information is more suitable for the Introduction, as it describes the mechanism of the laser.
Author Response
Reviewer 2.
The current study focuses on improving the lower tract urinary symptoms and female incontinence through non-invasive approaches. By boosting the vaginal collagen regeneration rate, the suspensory system of the urethra is strengthened, with significant impact upon the quality-of-life of the patients and minimal discomfort.
Overall, the paper is well put together. Regarding the use of English language, I would suggest some minor revisions to be done, as it can be improved.
Q1. The introduction describes concisely the current therapeutic options for the chosen pathology, as well as the novelty that Pixel CO2 Laser can bring in the treatment scheme of stress urinary incontinence. For this section, I would suggest for the authors to offer a clear definition of ‘’pixel’’ laser, as well as mentioning the present success versus failure rates and complication rates of Tension-Free sub-urethral slings.
Our reply: Thanks for the careful review, we’ve added a paragraph to further describe the definition of “pixel” in introduction as followed from line 61 to line 66:
“However, some research depicted that CO2 laser possesses the advantages, including tissue tightening fibroplasia per micrometer depth of damaged skin is more evident and lasts longer compared to Er: YAG laser [1,2]. Among the CO2 laser technique, the Pixel CO2 laser technology could pixelate the beam to either 49 or 81 pixels. The pixelated beams could generate micro-ablative damage by these microscopic columns. The deep thermal effect of Pixel applicator enable it caused the minimal ablation to patient and may ideal for SUI patients. The undamaged tissue led to a faster regeneration when the wound healing began.” and added the following sentence to mention the present success versus failure rates and complication rates of Tension-Free sub-urethral slings in introduction from line 49 to line 54 as followed:
“The golden standard operation for treating SUI has been Tension-free sub-urethral slings in the past decade [3]. Besides, tape insertion would be a solution when the symptoms come to moderate or severe, especially suitable for menopausal women [4]. However, the failure rate of Tension-free sub-urethral slings was reported in 5%–20% of patients [5] and the complication rate was also observed in 16% of patients [6].”
- Trelles, M.A.; Rigau, J.; Pardo, L.; García-Solana, L.; Vélez, M. Electron microscopy comparison of CO2 laser flash scanning and pulse technology one year after skin resurfacing. International journal of dermatology 1999, 38, 58-64, doi:10.1046/j.1365-4362.1999.00626.x.
- Ross, E.V.; McKinlay, J.R.; Anderson, R.R. Why does carbon dioxide resurfacing work? A review. Archives of dermatology 1999, 135, 444-454, doi:10.1001/archderm.135.4.444.
- Reisenauer, C.; Muche-Borowski, C.; Anthuber, C.; Finas, D.; Fink, T.; Gabriel, B.; Hübner, M.; Lobodasch, K.; Naumann, G.; Peschers, U.; et al. Interdisciplinary S2e Guideline for the Diagnosis and Treatment of Stress Urinary Incontinence in Women: Short version - AWMF Registry No. 015-005, July 2013. Geburtshilfe und Frauenheilkunde 2013, 73, 899-903, doi:10.1055/s-0033-1350871.
- Paz-Levy, D.; Weintraub, A.Y.; Reuven, Y.; Yohay, Z.; Idan, I.; Elharar, D.; Yohay, D. Prevalence and risk factors for urinary tract infection following stress urinary incontinence surgery with two midurethral sling procedures. 2018, 143, 333-338, doi:https://doi.org/10.1002/ijgo.12680.
- Kwon, J.; Kim, Y.; Kim, D.Y. Second-Line Surgical Management After Midurethral Sling Failure. International neurourology journal 2021, 25, 111-118, doi:10.5213/inj.2040278.139.
- Barisiene, M.; Cerniauskiene, A.; Matulevicius, A. Complications and their treatment after midurethral tape implantation using retropubic and transobturator approaches for treatment of female stress urinary incontinence. Wideochirurgia i inne techniki maloinwazyjne = Videosurgery and other miniinvasive techniques 2018, 13, 501-506, doi:10.5114/wiitm.2018.75871.
Q2: Regarding Materials and methods, the actual employed technique could be better and explained and further detailed. Additionally, intra-procedural pictures would be of great interest. Furthermore, this section could include a ‘definitions’ segment, where the authors should explain the urethral deviation and kinking angles and their normal values, as well as definitions for ‘positive pad test’ and if they measured the incontinence degree by assessing the number of used pads/day or by weighting each pad. Also, tables where each applied questionnaire could be detailed should be included.
Our reply: Thanks for your careful review, we’ve corrected the figure legends from line 110 to line 113 as followed:
“Figure 1. Bladder neck mobility from resting to straining. A, bladder neck at rest. B, bladder neck during straining; Figure 2. Urethral area- the hypoechoic area, marked in blue dotted line”
And added the segment from line 90 to line 93 as followed:
“The definitions of each test were listed below: positive pad test: The procedure of pad test followed the standard protocol from ICS for 1 hr pad test (pad weight >1g represent SUI patients [1]. The literature method used in our studies we used were listed included overactive bladder symptom score (OABSS) [2], UDI-6 , ICIQ-SF [3], IIQ-7[4], VLQ[5].”
- Krhut, J.; Zachoval, R.; Smith, P.P.; Rosier, P.F.; Valanský, L.; Martan, A.; Zvara, P. Pad weight testing in the evaluation of urinary incontinence. Neurourology and urodynamics 2014, 33, 507-510, doi:10.1002/nau.22436.
- Zhang, P.; Wu, Z.J.; Yang, Y. [Diagnosis of lower urinary tract voiding dysfunction with video-urodynamic studies]. Zhonghua Wai Ke Za Zhi 2010, 48, 1321-1324.
- Mikuš, M.; Ćorić, M.; Matak, L.; Škegro, B.; Vujić, G.; Banović, V. Validation of the UDI-6 and the ICIQ-UI SF - Croatian version. Int Urogynecol J 2020, 31, 2625-2630, doi:10.1007/s00192-020-04500-4.
- Momenimovahed, Z.; Tiznobaik, A.; Pakgohar, M.; Montazeri, A.; Taheri, S. Incontinence Impact Questionnaire (IIQ-7) and Urogenital Distress Inventory (UDI-6): Translation and Psychometric Validation of the Iranian Version. JOURNAL OF CLINICAL AND DIAGNOSTIC RESEARCH 2018, 12, doi:10.7860/JCDR/2018/34315.11538.
- Wilson, K.G.; Sandoz, E.K.; Kitchens, J.; Roberts, M. The Valued Living Questionnaire: Defining and measuring valued action within a behavioral framework. The Psychological Record 2010, 60, 249-272, doi:10.1007/BF03395706.
Q3: The Results highlight the main significant findings in clear and concise manner. Table 1 could detail the causes of urinary incontinence in each case. Moreover, I recommend to include more follow-up data from the failed cases, the presumed reason and the next therapeutic option for these women.
Our reply: Thanks for your valuable suggestion, we replenished the following paragraph in Material and methods from line 73 to line 76 as followed:
“The urinary incontinence cases in our studies usually caused by the problems of the muscle or nerves which contract the bladder and help the urine pass. Some cases suffered from SUI due to pregnancy and menopause. All patients in our studies were bladder neck hypomobility (type 2 SUI) and none of them was intrinsic sphincter deficiency (ISD, type 3 SUI).”. We also supplemented the description of SUI type in Table 1 as followed:
Besides, we have to apologize for the lack of follow-up data from the failed cases. We’ll try to make it complete in our future study.
Q4: Finally, the Discussions could benefit from adding a paragraph with the reported complications of Pixel CO2 laser in literature (Fibrosis? Cystitis?), as well as other uses for this type of laser therapy.
Our reply: Thanks for the careful review, we’ve added a paragraph in Discussions from line 169 to line 174 as followed:
“Although Pixel (fractional) CO2 laser technology patients got a lower rate of complications than non-fractionated ablative laser treatment, adverse effects might still happen with the best technology. Fractional CO2 laser were used in the aesthetic indication most such as post acne scarring [1]. Hence the most common complications of fractional CO2 laser reported were erythema (95.38%) and burning sensation (92.30%) after the procedure [2]. Some side effects included dryness and edema were seen in about 70% of patients [3]. The appropriate precautions should be perform during the procedure to avoid the possible side effects.“
- Petrov, A.; Pljakovska, V. Fractional Carbon Dioxide Laser in Treatment of Acne Scars. Open Access Macedonian Journal of Medical Sciences 2015, 4, doi:10.3889/oamjms.2016.004.
- Ross, R.B.; Spencer, J. Scarring and persistent erythema after fractionated ablative CO2 laser resurfacing. Journal of drugs in dermatology : JDD 2008, 7, 1072-1073.
- Salvatore, S.; Nappi, R.E.; Zerbinati, N.; Calligaro, A.; Ferrero, S.; Origoni, M.; Candiani, M.; Leone Roberti Maggiore, U. A 12-week treatment with fractional CO2 laser for vulvovaginal atrophy: a pilot study. Climacteric : the journal of the International Menopause Society 2014, 17, 363-369, doi:10.3109/13697137.2014.899347.
Q5: My final comment concerns the fifth paragraph of this section. The presented information is more suitable for the Introduction, as it describes the mechanism of the laser.
Our reply: Thanks for the valuable suggestion, we moved the paragraph to the introduction section as your suggestion.